# The Proteasome Inhibitor Bortezomib Induces Apoptosis and Activation in Gel-Filtered Human Platelets

**DOI:** 10.3390/ijms22168955

**Published:** 2021-08-19

**Authors:** Harriet Ghansah, Ildikó Beke Debreceni, Zsolt Fejes, Béla Nagy, János Kappelmayer

**Affiliations:** 1Department of Laboratory Medicine, Faculty of Medicine, University of Debrecen, H-4032 Debrecen, Hungary; harriet.ghansah@med.unideb.hu (H.G.); ideb@med.unideb.hu (I.B.D.); fejes.zsolt@med.unideb.hu (Z.F.); nagy.bela@med.unideb.hu (B.N.J.); 2Kálmán Laki Doctoral School, Faculty of Medicine, University of Debrecen, H-4032 Debrecen, Hungary

**Keywords:** proteasome inhibitor, human blood platelets, apoptosis, thrombin generation

## Abstract

Bortezomib (BTZ) has demonstrated its efficacy in several hematological disorders and has been associated with thrombocytopenia. There is controversy about the effect of BTZ on human platelets, so we set out to determine its effect on various types of platelet samples. Human platelets were investigated in platelet-rich plasma (PRP) and as gel-filtered platelets (GFPs). Mitochondrial inner membrane potential depolarization and phosphatidylserine (PS) and P-selectin expression levels were studied by flow cytometry, while thrombin generation was measured by a fluorescent method. In PRP, BTZ caused negligible PS expression after 60 min of treatment. However, in GFPs, PS expression was dose- and time-dependently increased in the BTZ-treated groups, as was P-selectin. The percentage of depolarized cells was also higher after BTZ pretreatment at both time points. Peak thrombin and velocity index increased significantly even with the lowest BTZ concentration (*p* = 0.0019; *p* = 0.0032) whereas time to peak and start tail parameters decreased (*p* = 0.0007; *p* = 0.0034). The difference between PRP and GFP results can be attributed to the presence of plasma proteins in PRP, as the PS-stimulating effect of BTZ could be attenuated by supplementing GFPs with purified human albumin. Overall, BTZ induces a procoagulant platelet phenotype in an experimental setting devoid of plasma proteins.

## 1. Introduction 

Proteasome, present in the nucleus and cytoplasm of eukaryotic cells, is responsible for the degradation of damaged, misfolded or unfolded proteins [1], regulating a variety of cellular pathways, including apoptosis, cell growth and proliferation, transcription, DNA repair, immune responses and signaling processes [2,3]. The most studied proteasome complex is the 26S proteasome consisting of a 20S catalytic core and one or two 19S regulatory subunits on either end of the 20S core. The 20S catalytic core is barrel-shaped, containing four heptameric rings arranged into two outer α-rings (α 1–7) and two inner β-rings (β 1–7) [4,5,6,7]. The alpha rings regulate the entry of the substrate into the 20S core and the interaction with the regulatory subunit. The beta rings constitute the proteolytic core where β1, β2 and β5 subunits have caspase-like, trypsin-like and chymotrypsin-like activities, respectively [5,8]. 

Proteasome inhibitors cause the accumulation of otherwise degradable proteins within the cell that eventually induces cell death [9,10]. The main target of proteasome inhibitors is the chymotrypsin-like β5 subunit of the proteasome [11]. Three proteasome inhibitors are currently in clinical use. Bortezomib (Velcade) was first approved by the United States Food and Drug Administration in 2003, followed by carfilzomib (Kyprolis) in 2012 and ixazomib (Ninlaro) in 2015 [12,13]. Others in clinical trials include marizomib (NPI-0052) [14,15], oprozomib (ONX-0912) [14,15] and delanzomib (CEP-18770) [14]. 

Bortezomib (BTZ) is a boronic acid dipeptide that binds reversibly not only to the chymotrypsin-like β5 site of the proteasome but also to the caspase-like β1 and trypsin-like β2 subunits at higher drug concentrations [16]. It has a molecular formula of C_19_H_25_BN_4_O_4_, and its chemical IUPAC name is [3-methyl-1-(3-phenyl-2-pyrazinylcarbonylamino-propanoyl)amino-butyl] boronic acid [17]. BTZ is a first-generation proteasome inhibitor used to treat newly diagnosed, relapsed, or refractory multiple myeloma and mantle cell lymphoma [3,18,19,20]. As a single drug and in combination with other agents, BTZ has also shown clinical efficacy in light-chain amyloidosis, Waldenstrom macroglobulinemia and peripheral T-cell lymphomas [21]. Several clinical trials on the efficacy of BTZ in other hematological diseases, such as acute lymphoid and myeloid leukemia, indolent B-cell non-Hodgkin lymphoma and diffuse large B-cell lymphoma, are ongoing [21]. The role of BTZ in immunosuppression has also been reported in patients with nonobstetric antiphospholipid syndrome [22]. 

Platelets, like nucleated cells, have an active proteasome system [23,24]. Earlier, a study by Ostrowska et al. showed that the human platelet 20S proteasome has a chymotryptic-like activity [25]. Subsequently, a global proteomic analysis has shown that platelets express all the three proteolytic activities of the proteasome: caspase-like, trypsin-like and chymotrypsin-like activities [26]. Although the role of platelet proteasome in protein degradation is not clear [23], its role in regulating platelet production [27] and lifespan [28] has been suggested.

BTZ is commonly known to induce thrombocytopenia in patients within a few days of starting treatment and is usually withdrawn if thrombocytopenia is severe. However, the platelet count recovers completely and rapidly between treatment cycles. Cases of thrombocytopenia-associated bleeding have been reported in patients treated with BTZ, and some patients required platelet transfusion [29]. A study has shown that the mechanism of BTZ-induced thrombocytopenia is independent of bone marrow injury or decreased thrombopoietin production [29]. Other studies have related this phenomenon mainly to inhibition of proplatelet formation of megakaryocytes, due to an increase in the levels of activated small GTPase Rho, a negative regulator of platelet formation [27,30]. A series of papers evaluated the dose-dependent inhibitory effect of BTZ in human platelet proteasomes and uncovered the effect of BTZ on platelet responsiveness and signaling [31,32,33].

To the best of our knowledge, the effect of BTZ in different types of platelet samples has not been studied in-depth. Although Nayak et al. demonstrated the regulatory role of proteasomal inhibition on platelet life span [28], their work did not focus on BTZ per se and was more related to animal models than humans. In view of this, we hypothesized that inhibition of human platelet proteasome activity by BTZ leads to platelet activation and results in a procoagulant platelet phenotype with subsequent thrombin generation. 

## 2. Results 

### 2.1. Platelet Activation and Apoptosis Studies with BTZ

To determine whether BTZ induces PS exposure in platelets, we used drug concentrations that have been described as steady-state concentrations in BTZ-treated patients [34]. First, PS exposure in human platelet-rich plasma (PRP) pretreated with BTZ (26 nM, 260 nM and 2.6 µM) and thrombin receptor-activating peptide (TRAP) was investigated by flow cytometry. After 60 min of treatment, BTZ elicited a nonsignificant PS exposure in PRP, while significantly induced PS expression was observed in TRAP-stimulated platelets (*p* < 0.0001) (Figure 1A). 

Since BTZ binds approximately 83% to human plasma proteins at therapeutic concentrations [35,36], its potential effect on plasma-free platelets was next determined by isolating platelets from plasma of healthy individuals via gel-filtration chromatography. These gel-filtered human platelets (GFPs) were treated with increasing BTZ concentrations similar to those used in the PRP. Here, thrombin was used as positive control. The representative dot plots show the effect of BTZ and thrombin on PS exposure to the platelet surface (Appendix A). Thrombin-treated platelets showed high PS exposure at 15 and 60 min of treatment. Nevertheless, BTZ pretreatment caused a similarly high and dose-dependent PS exposure at 15 min, leading to a double increase in PS over time by 60 min (Figure 1B). These results justified the activating effect of BTZ on isolated platelets; thus, we utilized GFPs in subsequent experiments. 

To confirm the possible effect of BTZ on platelet activation, we investigated platelet *α*-granule secretion by assessing surface P-selectin expression. As a result, human GFP samples were preincubated with BTZ and thrombin as indicated, and surface P-selectin expression was analyzed by flow cytometry. After 15 min incubation, a marked increase in P-selectin expression was observed with both agents, and these values further increased dose-dependently in the case of BTZ by 60 min (Figure 2).

To characterize the effect of BTZ on the regulation of platelet lifespan, subsequently, the depolarization of the platelet mitochondrial inner membrane potential (ΔΨm) was evaluated. Human GFPs were treated with increasing BTZ concentrations and thrombin for the indicated times, and changes in the ΔΨm were monitored using the cationic, cell-penetrating dye 3,3′-dihexyloxacarbocyanine iodide ((DiOC_6_(3)). Depolarization of ΔΨm was quantified as an increase in the percentage of depolarized cells and a decrease in the fluorescence of DiOC_6_(3)-stained platelets. Representative dot plots of the percentage depolarized cells of one of twelve experiments are shown (Appendix A). The percentage of depolarized cells in thrombin-treated platelets was significantly higher at the specified times, as was that in all the BTZ samples (Figure 3A). The median fluorescence intensity (MFI) of depolarized cells, on the other hand, decreased accordingly (Figure 3B). 

### 2.2. Functional Assays for Enhanced Phosphatidylserine (PS) Expression 

Since BTZ induces PS exposure in GFPs, the thrombin generation test was performed as a global evaluation of hemostasis. GFPs were treated with BTZ and TRAP for 60 min since this period resulted in a double increase in PS exposure. Subsequently, platelets were resuspended in autologous platelet-poor plasma (PPP) to a final concentration of 20 × 10^9^/L. Thrombin generation was initiated with 1 pM rTF. The kinetics of thrombin generation (lag time, time to peak and start tail) and the quantity of generated thrombin (peak thrombin and endogenous thrombin potential (ETP)) were studied. Characterized parameters of thrombin generation curve include (i) lag time, the moment at which thrombin generation starts (minutes); (ii) time to peak, the time until thrombin peaks (minutes); (iii) start tail, the time at which thrombin generation is completed (minutes); (iv) peak thrombin, the highest concentration of thrombin (nM); and (v) ETP, the area under the curve (nM × minutes).

The representative thrombin generation curves show the effect of BTZ on in vitro thrombin formation (Figure 4A). The thrombin peak value was significantly increased in all groups treated with BTZ and TRAP compared to the control group. Interestingly, BTZ at 26 nM elicited nearly the same thrombin peak as BTZ at 2.6 µM (Figure 4B). The ETP was significantly elevated after treatment with TRAP but not after BTZ pretreatment (Figure 4C). Although the lag time values were smaller, no significant changes were observed between BTZ-treated groups compared to the control group (Figure 4D). The time to peak (Figure 4E) and start tail (Figure 4F), however, were both significantly shorter compared to the control even at the low nanomolar BTZ concentration. Velocity index, a calculated parameter of the thrombin generation defined as peak thrombin/time to peak-lag time, was also significantly elevated in the BTZ-treated samples (Appendix A).

During activation or apoptosis, platelets release microparticles through cellular blebbing [37]. Platelet-derived microparticles (PMPs) express PS, which also provides binding sites for prothrombinase complexes (FXa/Fva), thus promoting thrombin formation. In this regard, we investigated the possible role of BTZ in inducing PMP formation. All BTZ-treated groups showed a significantly higher number of PMPs compared to the control group (Figure 5). TRAP showed only a weak stimulatory effect on PMP formation as compared to control which is consistent with the results of a previous study [38].

### 2.3. Plasma Proteins Neutralize the Effect of BTZ on Human Platelets In Vitro

We intended to demonstrate in vitro that plasma proteins exert an inhibitory effect on BTZ; thus, GFPs were either supplemented with human serum albumin (HSA) or buffer prior to treatment with BTZ and thrombin. Albumin was used in this study as the most abundant plasma protein. For PS expression, pretreatment with BTZ was 60 min, since we found that BTZ did not cause significant PS expression in PRP at this time, unlike GFPs. As expected, PS and P-selectin expression levels were increased in platelets without the addition of albumin. However, enhanced expression of PS and P-selectin was prevented by albumin, even after preincubation with subnormal albumin concentrations (Figure 6A,B).

With 1 U/mL thrombin as positive control in this study, we did not observe changes in PS expression in GFPs with or without albumin addition: PS expression levels were 32 ± 7.6%, 33 ± 6.6% and 31 ± 7.5% in the presence of 10, 20 and 40 mg/mL of albumin, respectively, compared to the non-albumin group where PS expression was 31 ± 5.7%. Similarly, the addition of albumin did not affect thrombin-induced P-selectin expression, which was 95 ± 2.6% vs. 94.7 ± 2.5% in albumin and non-albumin groups (data not shown).

## 3. Discussion

The proteasome inhibitor BTZ demonstrates antiproliferative and antitumor activities via inhibiting the proteasomal degradation of numerous regulatory proteins after ubiquitination. This drug has proven efficacy in several malignant hematological disorders, and it has primarily revolutionized the treatment of refractory and relapsed multiple myeloma. BTZ has demonstrated superiority to high-dose dexamethasone treatment, resulting in significantly more patients achieving complete or partial remission [39]. Subsequently, it was also approved for previously untreated patients with multiple myeloma or mantle cell lymphoma. BTZ acts in the bone marrow microenvironment by inhibiting the binding of myeloma cells to the stromal cells. This may be achieved since malignant plasma cells are more dependent on proteasomal activity likely because of the higher protein turnover and thus being more sensitive to its blockage [40]. It is also suspected that aside from the main site of action, BTZ exerts an effect on several other cell types. In preliminary experiments, we verified the apoptosis-inducing effect of BTZ on magnetically isolated peripheral blood B-cells of a healthy donor. In line with the results of a previous study [41], an increase in PS exposure was observed in all BTZ-treated samples compared to the control sample (Appendix A). 

In this study, we intended to investigate peripheral blood platelets in two different sample types. The generally used platelet sample in numerous former experiments was the PRP. This sample can conveniently be prepared and is useful for numerous platelet studies including platelet aggregometry, the gold-standard platelet function test and platelet-dependent thrombin generation. We hypothesized that platelets will be activated by BTZ preincubation and this will generate extracellular PS expression that may lead to enhanced thrombin generation. However, we only experienced a nonsignificant elevation in platelet PS expression, and this did not cause any significant change in thrombin generation (data not shown).

We thought that since BTZ is highly protein-bound in the circulation [35,36], consequently plasma proteins could neutralize its effect on platelets in vitro. Thus, we intended to use platelets devoid of plasma proteins. In preliminary experiments, we used both washed platelets and GFPs and have established that gel-filtration is a more gentle way of platelet preparation that results in much lower ‘artificial’ P-selectin expression. By measuring protein concentration by the Pierce BCA protein assay, we could verify that the GFPs used in this study were completely devoid of plasma proteins (Appendix A). Our results were in accordance with those of Nayak et al. who demonstrated in washed platelets that in mice both 0.1 mg/kg and 0.3 mg/kg dose of BTZ results in significant elevation of annexin V binding [28]. Similarly, in our experiments, we determined PS expression by annexin V binding in GFPs and found that already after 15 min, BTZ induces significant PS expression that further increases at 60 min. However, in our experimental settings, we used much lower doses of BTZ where this effect can be observed in the low nanomolar concentration range that is considered as low therapeutic plasma concentration in treated myeloma patients. Similar to PS expression, the ΔΨm depolarization was also time-dependently increased and provided a stronger stimulus than the positive control with thrombin. In platelets, two mitochondrial mechanisms can lead to cell death [42], and only one of them evokes a procoagulant phenotype with simultaneous P-selectin activation. 

We wanted to prove that the expressed PS could indeed facilitate thrombin generation. Thus, we initiated thrombin generation by adding 1 pM of tissue factor only, and we could detect a shortening in the time parameters. After BTZ pretreatment, these differences became highly significant in the time to peak and start tail parameters. Furthermore, the peak thrombin concentration was significantly elevated already even at the lowest BTZ concentration. Since the peak thrombin values of applied BTZ concentrations usually surpassed that of a high concentration of the positive control TRAP, we thought that in addition to platelets, the thrombin generation is also facilitated by the PMPs in these samples. By using calibrated beads and adjusting the cursor upper threshold at the 1 µm bead, we could identify the percentages of events in this microparticle region that was considerably higher in the BTZ preincubated samples compared to those preincubated with TRAP. Thus, it is quite likely that the larger PMP number also contributed to the intense thrombin peak. This is conceivable as PMPs intensely express PS on their surface, which can promote thrombin formation. These data may have pathological significance since thrombotic events are frequently observed in multiple myeloma patients [43,44] and, in some cases, in association with proteasome inhibitor treatment [45]. In our study, the absolute number of PMPs between control and TRAP-treated platelets normalized for platelet number exactly matches that described by Takano et al. [38]. However, the result in our case was significantly higher compared to the negative control value, which may have been due to the larger number of experiments performed. We suggest that PMP formation and ΔΨm depolarization are sensitive parameters as they already peak at the lowest concentration of BTZ. The scope of our work did not include the determination of ubiquitinated protein content or proteasome activity. However, Koessler et al. previously showed that BTZ in the 10 nM to 10 µM range dose-dependently inhibits the proteasome activity in human platelets [31]. 

The discrepancy of low PS expression in PRP samples and significantly higher PS expression in GFPs could be explained by supplementing GFP buffer with purified human serum albumin before BTZ pretreatment. Albumin-supplemented GFPs failed to respond to BTZ and resulted in a modest PS elevation similarly to PRP. Even a very low albumin concentration (10 mg/mL) was sufficient to prevent PS expression of GFPs by 70%, which demonstrates the neutralizing effect of plasma albumin on the effect of BTZ. However, for the complete picture, it should be considered that targeted cells in other anatomical localizations such as in the bone marrow microenvironment are surrounded by a different protein milieu where the action of BTZ can be better exerted.

During in vivo thrombus formation, the contracting platelets translocate procoagulant platelets to the thrombus surface [46]. Furthermore, it was demonstrated in dual-activated platelets that the generation of surface PS expression is delayed compared to glycoprotein IIb/IIIa activation [47]. Thus, these events occur in a spatiotemporally and contextually regulated manner. These observations may also have important pharmacological consequences as the specific inhibition of the mitochondrial calcium entry might abrogate procoagulant platelet formation [48]. 

The PS-exposing peripheral blood cells can contribute to the thrombotic processes observed in a large percentage of multiple myeloma patients. PS-expressing red blood cells, leukocytes and platelets were detectable both in untreated patients and more so in patients on immunomodulatory drugs [49]. 

In conclusion, our studies provide further insight into the mechanism of BTZ action on platelets. GFPs are an ideal sample type due to their low degree of activation after the isolation procedure. In these samples, we could verify that platelets intensely respond to the presence of BTZ by generating procoagulant platelets via mitochondrial membrane depolarization, PS expression and microparticle formation, which can result in enhanced thrombin generation. However, it is important that these effects are attenuated even at low plasma albumin concentrations, and in PRP of BTZ-treated patients, the harsh effect of this drug on human platelets is largely blunted by albumin.

## 4. Materials and Methods

### 4.1. Materials 

BTZ (Velcade) was purchased from Selleckchem (Munich, Germany), dissolved in dimethyl sulfoxide (DMSO, Sigma-Aldrich, St. Louis, MO, USA) and stored at −20 °C until use. The following materials were also purchased from Sigma-Aldrich (St. Louis, MO, USA): Sepharose CL-2B column, human serum albumin (HSA), thrombin receptor-activating peptide (TRAP), bovine thrombin, 3,3′-dihexyloxacarbocyanine iodide ((DiOC_6_(3)) and paraformaldehyde (PFA). Anti-CD41-PE antibody was from DAKO (Glostrup, Denmark). Annexin V-FITC, annexin V-binding buffer, anti-CD42a-FITC, mouse IgG1-PE and anti-CD62P-PE antibodies were obtained from Becton Dickinson (San Jose, CA, USA). The size calibration beads (Megamix-Plus FSC) were purchased from Biocytex (Marseille, France). The following materials were used in the thrombin generation test: PRP reagent, calibrator, Fluo-Buffer, fluorogenic substrate (Thrombinoscope BV, Maastricht, The Netherlands) and a 96-well black microplate (Greiner Bio, One North America Inc., Monroe, MI, USA). 

### 4.2. Blood Collection

Peripheral venous blood was obtained with informed consent from a total of 41 healthy volunteer participants who had not taken any drugs that could affect platelet function in the last two weeks prior to sampling. Peripheral blood was drawn into tubes containing 3.2% (0.105 M) sodium citrate and processed without delay. Ethical approval was obtained from the Ethics Committee of the University of Debrecen (identification code: RKEB/IKEB 4875-2017, approval date: 25 September 2017). 

### 4.3. Plasma Preparation

Citrated whole blood from healthy subjects was centrifuged at 170 g for 15 min at room temperature (RT) to obtain PRP. Platelet counts were measured with a Sysmex XP-300 hematology analyzer (Sysmex, Kobe, Japan) and adjusted to 250 × 10^9^/L with PPP, prepared by centrifugation of a fraction of the PRP at 1500 g for 15 min at RT. For the thrombin generation test, PPP was prepared by double centrifugation. First, citrated whole blood was centrifuged at 1500× *g* for 15 min, and the plasma was collected and recentrifuged at 10,000× *g* for 10 min to remove all residual platelets. 

### 4.4. Isolation of Platelets by Gel-Filtration Chromatography 

Citrated whole blood from healthy subjects was diluted with equal volumes of buffered saline glucose citrate (BSGC, 129 mM NaCl, 1.6 mM KH_2_PO_4_, 14 mM Na citrate, 11 mM glucose and 10 mM NaH_2_PO_4_; pH 7.3) in plastic tubes and immediately centrifuged at 170 g for 15 min at RT to obtain PRP. Gel-filtration was performed on a Sepharose CL-2B column equilibrated with BSGC, and this same buffer was used to elute platelets. The eluates were collected into Eppendorf tubes, and the platelet counts were determined by the Sysmex XP-300 hematology analyzer. Platelets devoid of plasma proteins (determined by the Pierce BCA protein assay) were subsequently used in experiments. 

#### 4.4.1. PS Expression in Platelets and PMPs 

Platelet PS expression was measured with a flow cytometer by annexin V-binding. Human PRP and GFPs were incubated with BTZ (26 nM, 260 nM and 2.6 µM), thrombin (1 U/mL) or TRAP (40 µM) and DMSO (0.2%) for 15 and 60 min at 37 °C. To determine whether the effect of BTZ on human platelets is neutralized by plasma proteins in vitro, GFPs were supplemented with HSA dissolved in BSGC at concentrations of 10, 20 and 40 mg/mL at RT before 60 min treatment with BTZ at 37 °C. In all experiments, 5 µL of pretreated sample was stained with 5 µL each of annexin V-FITC and anti-CD41-PE antibody in 35 µL of annexin V-binding buffer (2.5 mM CaCl_2_). The samples were incubated for 15 min at RT in the dark, diluted to 550 µL with annexin V-binding buffer and immediately analyzed using an FC 500 flow cytometer (Beckman Coulter, Brea, CA, USA). Ten thousand platelets were acquired per sample and analyzed using the Kaluza software (Beckman Coulter, Brea, CA, USA). The PMP gate was set using size calibration beads (0.3–1 µm) and PMP analysis was done from tubes of PS-stained GFP samples. 

#### 4.4.2. P-Selectin Expression

Human GFPs were treated with BTZ (26 nM, 260 nM and 2.6 µM), thrombin (1 U/mL) and DMSO (0.2%) for 15 and 60 min at 37 °C. In some experiments, GFPs were supplemented with 40 mg/mL HSA before 15 min incubation with BTZ. The samples were fixed with 1% PFA, pH 7.4, for 1 h at RT in the dark followed by washing with phosphate-buffered saline (PBS) at 2500 g for 15 min. Forty microliters of washed resuspended platelets were stained with 5 µL each of anti-CD42a-FITC and anti-CD62P-PE antibodies. The samples were again washed twice and resuspended in PBS for flow cytometric analysis. In each experiment, 10,000 platelets were acquired per sample and analyzed using the Kaluza software.

#### 4.4.3. Mitochondrial Inner Membrane Potential (Δψm) Depolarization

Five microliters of GFPs treated with BTZ (26 nM, 260 nM and 2.6 µM), thrombin (1 U/mL) or DMSO (0.2%) was diluted 1:20 with BSGC and stained with 20 µL of 1.5 µM DiOC_6_(3) dye for 20 min at RT in the dark. The samples were diluted to 560 µL with BSGC, and platelets were acquired based on FSC and SSC properties. Fluorescence of DiOC_6_(3)-stained platelets was analyzed on the SSC-FL1 dot plot. The percentage of depolarized platelets was determined as a decrease in fluorescence of the DiOC_6_(3)-stained platelets. 

### 4.5. Thrombin Generation Assay

Human GFPs were treated with BTZ (26 nM, 260 nM and 2.6 µM), TRAP (40 µM) and DMSO (0.2%) for 60 min at 37 °C. The pretreated platelets were resuspended in autologous PPP to a final concentration of 20 × 10^9^/L, which was used in the thrombin generation test. Thrombin formation was measured with the Fluoroscan Ascent FL fluorimeter. Briefly, 20 µL of PRP reagent (1 pM recombinant tissue factor (rTF)) or calibrator were pipetted into a 96-well black microplate. Eighty microliters of samples were added and incubated for 10 min at 37 °C, and thrombin generation was started by the addition of 20 µL FluCa (Fluo-Buffer (contains CaCl_2_) and fluorogenic substrate (contains aminomethyl coumarin)). The following thrombin generation parameters were evaluated using the Thrombinoscope software (Thrombinoscope BV, Maastricht, The Netherlands); the lag time, peak thrombin, time to peak, endogenous thrombin potential (ETP) and start tail. 

### 4.6. Statistical Analysis

Data were analyzed using GraphPad Prism (GraphPad Software, San Diego, CA, USA). The test for normality was performed by the Kolmogorov–Smirnov test. For data with normal distribution, one-way analysis of variance (one-way ANOVA) was used to compare the differences between groups, with Dunnett’s test as post hoc analysis. The Kruskal–Wallis test was applied for data without normal distribution, followed by Dunn’s multiple comparisons test. For all comparisons, a *p*-value < 0.05 was considered statistically significant. 

## Figures and Tables

**Figure 1 ijms-22-08955-f001:**
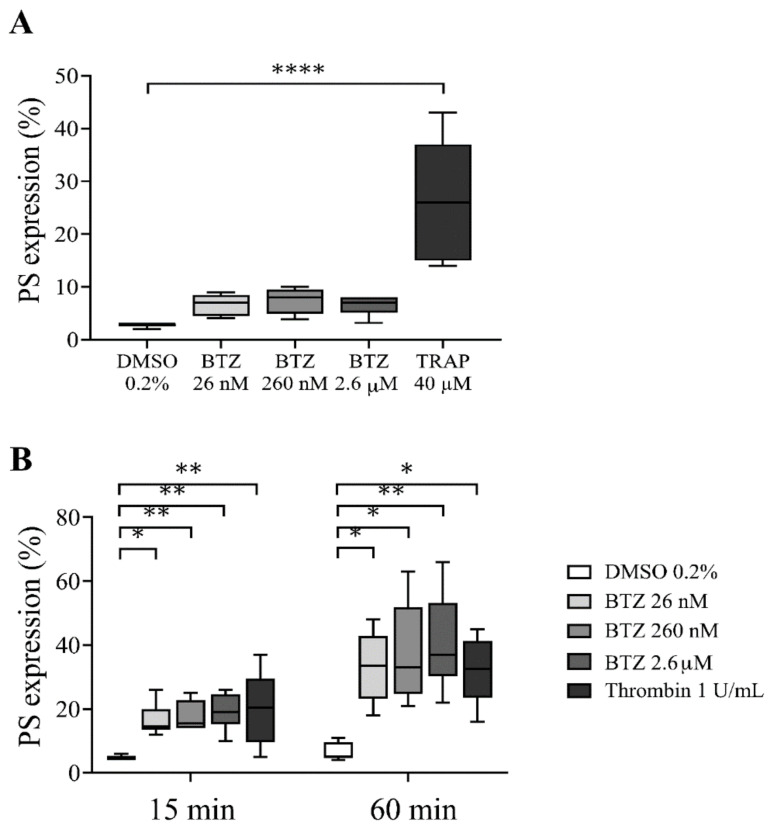
PS expression in BTZ-pretreated human PRP and GFPs. In panel (**A**), the PRP of healthy subjects was treated with BTZ, TRAP (positive control) and DMSO (negative control) for 60 min. In panel (**B**), platelets were isolated from the PRP of healthy subjects by gel-filtration and treated with BTZ, thrombin (positive control) and DMSO (negative control) for 15 and 60 min. PS expression was measured by flow cytometry based on CD41 and annexin V double positivity. Data are expressed as median and interquartile range of 5 experiments for panel (**A**) and 6 experiments for panel (**B**). The level of significance in panel (**A**) was determined using the Kruskal–Wallis test and Dunn’s multiple comparisons test; **** *p* < 0.0001 compared to the DMSO group. In panel (**B**), the level of significance at 15 min of treatment was determined using ANOVA and Dunnett’s multiple comparisons test; * *p* < 0.05 and ** *p* < 0.01 compared to the DMSO group. The level of significance at 60 min of treatment was determined using Kruskal–Wallis test and Dunn’s multiple comparisons test; * *p* < 0.05 and ** *p* < 0.01 compared to the DMSO group. BTZ: bortezomib; DMSO: dimethyl sulfoxide; PS: phosphatidylserine; TRAP: thrombin receptor-activating peptide.

**Figure 2 ijms-22-08955-f002:**
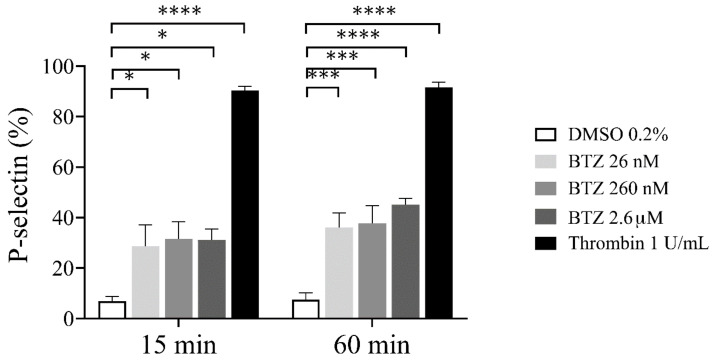
P-selectin expression in BTZ-preincubated human GFPs. GFPs were treated with BTZ, thrombin (positive control) and DMSO (negative control) for 15 and 60 min. P-selectin expression was analyzed by flow cytometry using anti-CD62P antibody. Mean ± SEM of 5 independent experiments is presented. The level of significance at both 15 and 60 min of treatment was determined using ANOVA and Dunnett’s multiple comparisons test; * *p* < 0.05, *** *p* < 0.001 and **** *p* < 0.0001 compared to the DMSO group. BTZ: bortezomib; DMSO: dimethyl sulfoxide.

**Figure 3 ijms-22-08955-f003:**
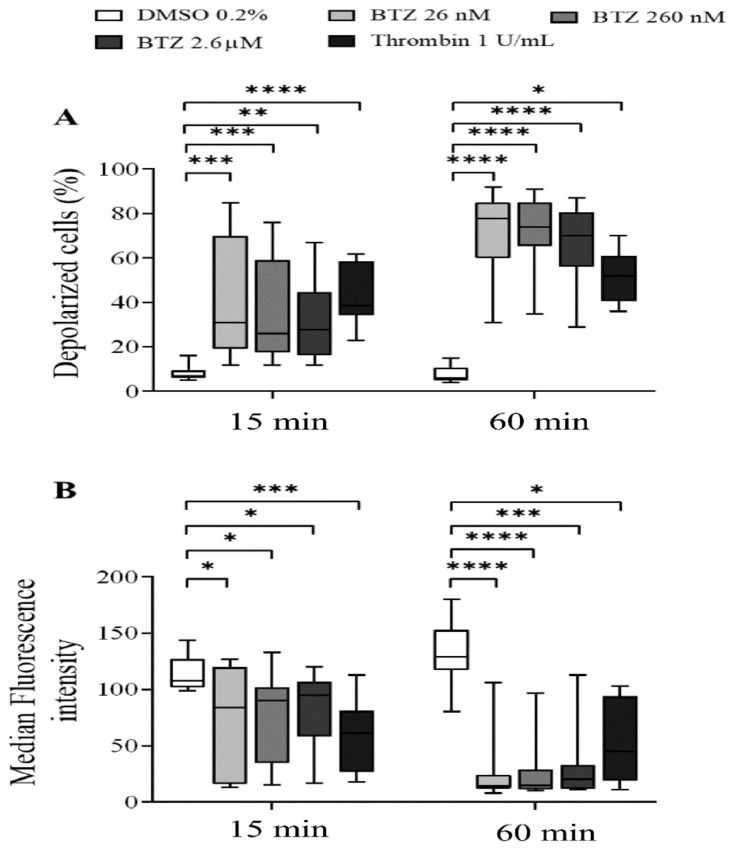
BTZ induces ΔΨm depolarization of human GFPs. GFPs were incubated with BTZ, thrombin (positive control) and DMSO (negative control) for 15 and 60 min. Depolarization of ΔΨm was analyzed by flow cytometry using DiOC_6_(3) dye. Data are expressed as median and interquartile range of the percentage of depolarized cells (**A**) and the median fluorescence intensity (**B**). The level of significance at both 15 and 60 min of treatment (*n* = 12) was determined using the Kruskal–Wallis test and Dunn’s multiple comparisons test; * *p* < 0.05, ** *p* < 0.01, *** *p* < 0.001 and **** *p* < 0.0001 compared to the DMSO group. BTZ: bortezomib; DMSO: dimethyl sulfoxide.

**Figure 4 ijms-22-08955-f004:**
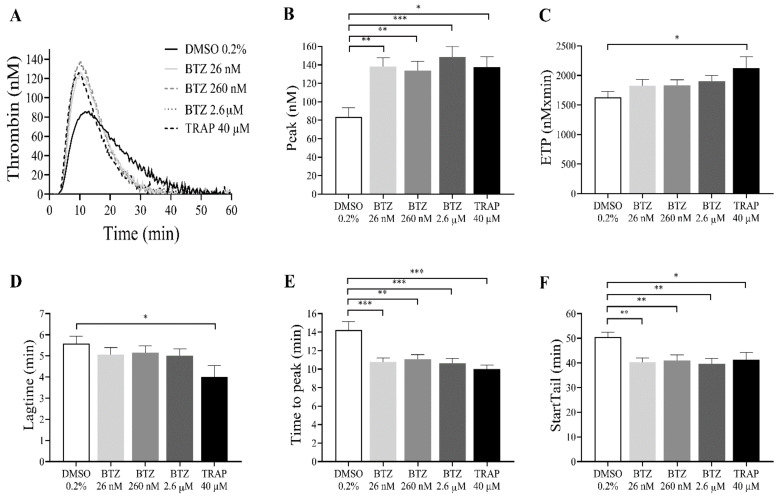
BTZ enhances in vitro thrombin formation in human GFPs. GFPs were treated with BTZ, TRAP (positive control) and DMSO (negative control) for 60 min and resuspended in autologous PPP to a final concentration of 20 × 10^9^/L. Thrombin generation was initiated with 1 pM rTF (PRP reagent). Representative thrombin generation curves of 1 of 10 independent experiments are shown (**A**). Mean ± SEM of 5 experiments for TRAP or 10 experiments for DMSO and BTZ is presented (**B**–**F**). The level of significance was determined using ANOVA and Dunnett’s multiple comparisons test; * *p* < 0.05, ** *p* < 0.01, and *** *p* < 0.001 compared to the DMSO group. BTZ: bortezomib; ETP: endogenous thrombin potential; DMSO: dimethyl sulfoxide; TRAP: thrombin receptor-activating peptide.

**Figure 5 ijms-22-08955-f005:**
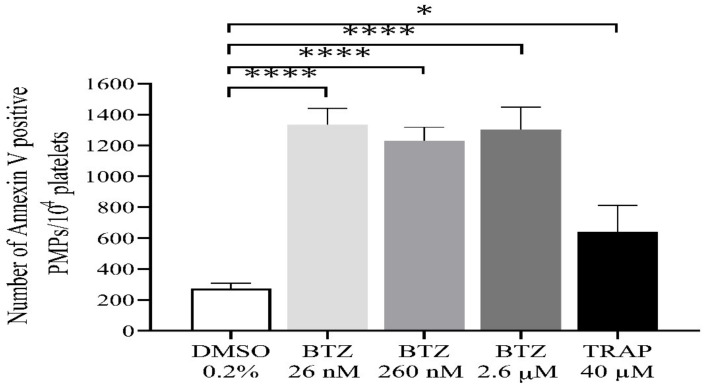
Detection of PMPs in BTZ-pretreated human GFPs. GFPs were treated with BTZ, TRAP (positive control) and DMSO (negative control) for 60 min. PMPs were analyzed by flow cytometry. The PMP gate was set using size (0.3–1 µm) calibration beads (Megamix-Plus FSC). Mean ± SEM of 5 experiments for TRAP or 10 experiments for DMSO and BTZ is presented. The level of significance was determined using ANOVA and Dunnett’s multiple comparisons test; * *p* < 0.05 and **** *p* < 0.0001 compared to the DMSO group. BTZ: bortezomib; DMSO: dimethyl sulfoxide; PMPs: platelet-derived microparticles; TRAP: thrombin receptor-activating peptide.

**Figure 6 ijms-22-08955-f006:**
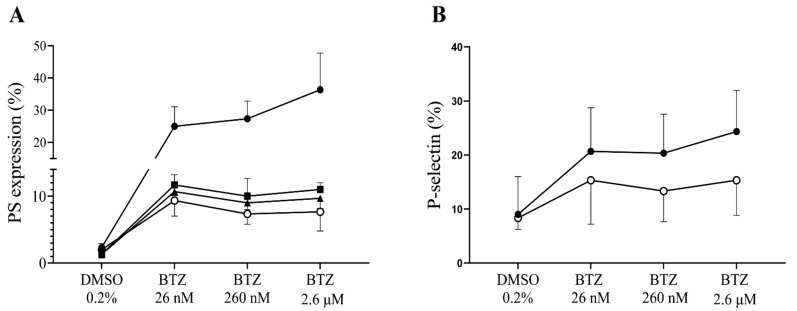
The effect of BTZ on human GFPs is neutralized by HSA. In panel (**A**), GFPs were supplemented with 10, 20 and 40 mg/mL of HSA or buffer before incubation with BTZ for 60 min. In panel B, 40 mg/mL of HSA was used and pretreatment with BTZ was performed for 15 min. Platelet PS and P-selectin expression levels were analyzed by flow cytometry using annexin V and anti-CD62P antibody, respectively. The expression of PS (**A**) and P-selectin (**B**) was induced in the BTZ-treated platelets without albumin (represented by line with solid circle). However, these effects were abolished with the addition of albumin (10, 20 and 40 mg/mL; represented by lines with square, triangle and open circle, respectively). Mean ± SD of 3 independent experiments is presented. BTZ: bortezomib; DMSO: dimethyl sulfoxide; PS: phosphatidylserine.

## Data Availability

The data presented in this study are available on request from the corresponding author.

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
