# Peer review of "The Proteasome Inhibitor Bortezomib Induces Apoptosis and Activation in Gel-Filtered Human Platelets"

_ijms, 2021, doi:10.3390/ijms22168955_

Round 1

Reviewer 1 Report

Ghansah et al. investigated the potential mechanism of bortezomib induced thrombocytopenia, which is the main drawback for its use for MM. There is a major flaw in this manuscript. Authors present a good portion of evidence demonstrating the procoagulant potential of bortezomib treated platelets and also referring to them as apoptotic. Whereas apoptotic platelets are not procoagulant, rather platelets dying with necrotic pattern demonstrate procoagulant functionality. Kholmukhamedov et al. Thrombosis and haemostasis 117 (11), 2207-2208 extensively discusses this. Also,  Kholmukhamedov et al. Blood 132, 2420 have shown that only necrotic and not apoptotic platelets support active prothrombinase.  

All the data presented in this manuscript including PS exposure, support in thrombin generation, etc. is indicative that bortezomib induces procoagulant platelet formation. It has to be mentioned, however, that support of coagulation by procoagulant platelets is spatio-temporally and contextually regulated. Besides this there are a couple of other points authors need to work on: 

  1. Was corn trypsin or any other inhibitor for contact activation added during thrombin generation assay? If so, state it. 
  2. Line 130 states “crucial feature of apoptosis is the loss of mitochondrial membrane potential” which is not true, apoptosis is an energy requiring process and therefore disruption of MMP means there is virtually no ATP production and therefore cells with complete MMP disruption cannot undergo cell death via apoptosis.
  3. Line 388 - G/L is not a common abbreviation, please correct. 

Author Response

Response to Reviewer 1.

We would like to thank the Reviewer for the comments and the following responses are provided. Our manuscript has been considerably revised as several other Reviewers raised similar points.

We paid special attention to the parts where the Reviewer indicated „Must be improved”. The platelet apoptosis versus necrosis issue has been extensively rephrased. We included many additional manuscripts in the References including the ones suggested by the Reviewer. Based on these considerations, the manuscript was upgraded and the following new part was inserted.

Lines 303-305:

In platelets two mitochondrial mechanisms can lead to cell death [42] and only one of them evokes a procoagulant phenotype with simultaneous P-selectin activation.

Lines 351-357:

During in vivo thrombus formation, the contracting platelets translocate procoagulant platelets to the thrombus surface [46]. Furthermore, it was demonstrated in dual activated platelets, that the generation of surface PS expression is delayed compared to glycoprotein IIb/IIIa acitivation [47]. Thus these events occur in a spatio-temporally and contextually regulated manner. These observations may also have important pharmacological consequences as the specific inhibition of the mitochondrial calcium entry might abrogate procoagulant platelet formation [48]

Numbered questions:

  1. Corn trypsin inhibitor was not used in our experiments as we did not want to block the intrinsic pathway of the coagulation cascade. It has recently been shown that several activators and modulators can exert their effect via the contact system. Also, in our experiments, we detected P-selectin expression (alpha-granule release). Quite similarly, dense granules can release their cargo into the surrounding, one of which is polyphosphate. Polyphosphate recently emerged as a hemostatic modulator and due to its highly anionic nature, it is considered a natural activator of the contact system. Thus, the complete effect on thrombin generation would have been missed if contact system had been blocked.
  2. The Reviewer is right and we considerably rephrased this part. The revised version is also in accordance with recent talks delivered at the 2021 online ISTH meeting, where procoagulant and apoptotic platelets are distinguished. Obviously, we identified procoagulant platelets but we are aware that the PRP and GFP samples that we investigated are simple models, and during in vivo thrombus formation, these processes happen in a heterogenous thrombus where PS-exposing platelets are squeezed to the thrombus surface by the larger group of clot retracting platelets (Nechipurenko et al, ATVB, 2018).

In order to see that our BTZ treated samples undergo cell lysis (necrosis), in additional experiments we performed LDH measurements from the supernatants of the BTZ-pretreated GFP and compared to results of a sonicated positive controls as LDH percentage of total (lysed cells) (Figure 1 A). We also carried out this experiment for another proteasome inhibitor carfilzomib (CFZ) and found similar results (Figure 1 B). This refers to cell lysis in case of both proteasome inhibitors.

  1. We replaced the G/L to the more widely used term 109/L.

Reviewer 2 Report

This manuscript reports that the proteasome inhibitor bortezomib (BTZ) induces, at low nanomolar concentrations, apoptotic processes in human platelets using PRP or GFP. As parameters, PS expression in platelets and PMPs,  P-selectin expression, mitochondrial inner membrane potential (ΔΨm) depolarization and thrombin generation assay were used.

Overall, the experiments are well done and described. However, the novelty of results is limited, and there are some other limitations.

  • The introduction and the discussion of the paper only partly reflects the knowledge in the field. There are a number of papers in the literature which report short- and long-term effects of bortezomib on various aspects of platelet activation or inhibition. The authors should do a better job in integrating this knowledge in their present paper.
  • This paper relies on one pharmacological compound. However, other proteasome inhibitors are there/in development. The authors should perhaps study (as some kind of functional/positive control) another proteasome inhibitor.
  • The authors only used functional read-outs for the analysis. Actually, it is possible to measure the primary read-out biochemically, inhibition of the proteasome activity, accumulation of poly-ubiquitylated proteins). Were such studies done?
  • Finally, the still unanswered/ open questions and future tasks should be better addressed in the discussion.

Author Response

Response to Reviewer 2.

  1. We agree with this comment and other Reviewers also raised this issue. The revised version of our manuscript is now completed with seven new references that are either relevant to the effect of bortezomib on platelets or the fine differentiation of apoptotic and necrotic platelets.

  1. We focused on the effect of bortezomib on platelets and as a positive control we utilized PAR receptor agonists. Nevertheless, we agree with the Reviewer that there are other proteasome inhibitors and one of them is increasingly used in clinical practice. Thus, we designed experiments where gel-filtered platelets (GFP) were pretreated with carfilzomib (CFZ). The therapeutically relevant CFZ concentrations (Maruyama D et al. Cancer Science. 2018, 109:3245-3252) were used in these experiments. We did magnetic separation of B-cells as described in the Materials and Methods and incubated them with different concentrations of CFZ for 24 hours at 37 °C and 5% CO2 (see enclosed Figure 1 for the Reviewer). We also carried out similar experiments with GFP. To our surprise, even lower concentrations of CFZ (the lowest of the therapeutic range) elicited PS expression and platelet-derived microparticle in GFP (see enclosed Figures 2 and 3 for the Reviewer). Overall, these preliminary data prove that other proteasome inhibitors may elicit similar effects. We do not want to include these data in the manuscript as the time to prepare the revised version is really short and only a limited number of experiments could be done. Nevertheless, we are grateful for this idea that may be better outlined in future studies.

  1. In the limited time-frame to prepare the revised version, we had no chance of setting up new assay for determining ubiquitinated protein content or proteasome activity. Nevertheless, we agree with the Reviewer that all our tests are indirect proof of proteasome inhibition. Thus, we rephrased the Discussion and cited the relevant papers that in the past years clearly showed that bortezomib in the 10 nM -10 uM range dose-dependently inhibits the proteasome activity in human platelets (Koessler J et al. Eur J Pharmacol. 2016, 791:99-104.) These authors also found that a fluorescent assay to detect proteasome activity is more sensitive than the measurement of polyubiquitinated protein content.

  1. We rephrased the Discussion according to the Reviewer’s suggestions and inserted the following part to lines: 303-321.

During in vivo thrombus formation, the contracting platelets translocate procoagulant platelets to the thrombus surface [46]. Furthermore, it was demonstrated in dual activated platelets, that the generation of surface PS expression is delayed compared to glycoprotein IIb/IIIa acitivation [47]. Thus these events occur in a spatio-temporally and contextually regulated manner. These observations may also have important pharmacological consequences as the specific inhibition of the mitochondrial calcium entry might abrogate procoagulant platelet formation [48].

The PS exposing peripheral blood cells can contribute to the thrombotic processes observed in a large percentage of multiple myeloma patients. PS expressing red blood cells, leukocytes and platelets were detectable both in untreated patients and more so in patients on immunomodulatory drugs [49].

In conclusion, our studies provide further insight into the mechanism of BTZ action on platelets. GFP are an ideal sample type due to its low degree of activation after isolation procedure. In these samples we could verify that platelets intensely respond to the presence of BTZ by generating procoagulant platelets via mitochondrial membrane depolarization, PS expression and microparticle formation, that can result in enhanced thrombin generation. However, it is important that these effects are attenuated even at low plasma albumin concentrations, and in PRP of BTZ-treated patients, the harsh effect of this drug on human platelets is largely blunted by albumin.

Reviewer 3 Report

The manuscript by Ghansah et al. describes the behaviour of gel-filtered platelets and platelets from platelet-rich-plasma after exposure to bortezomib.
The paper is well-organized and written in good English. The results are relevant for the readers of IJMS, although it would be useful to further stress the novelty of the performed experiments in comparison with previously published manuscripts (i.e. ref. 31).
I suggest the addition of a brief conclusion section highlighting the clinical impact of the reported findings.

Author Response

Response to Reviewer 3.

We would like to thank the Reviewer for the positive feedback. Based on the comments of all four Reviewers we have considerably rephrased our manuscript and also more clearly described the novelty of our findings.

As can be seen in the track changes version of our revised manuscript, we improved several parts of the manuscript that also better highlights the novelty of our findings.

The following conclusion parts have been inserted into the discussion in lines 314 to 321 of the revised manuscript.

In conclusion, our studies provide further insight into the mechanism of BTZ action on platelets. GFPs are an ideal sample type due to its low degree of activation after isolation procedure. In these samples we could verify that platelets intensely respond to the presence of BTZ by generating procoagulant platelets via mitochondrial membrane depolarization, PS expression and microparticle formation, that can result in enhanced thrombin generation. However, it is important that these effects are attenuated even at low plasma albumin concentrations, thus in PRP of BTZ-treated patients, the harsh effect of this drug on human platelets is largely blunted by albumin.

Reviewer 4 Report

COMMENTS TO AUTHORS:

The manuscript describes the influence of proteasome inhibitor bortezomib (BTZ) on human platelets. The aim of this study is interesting and is a good complement to knowledge about the interaction and role of proteasome in platelet activity and lifespan. The advantage of this study is the comparison of bortezomib activity towards the isolated platelets suspended in buffer only or the isolated platelets suspended in buffer supplemented with plasma protein (human albumin) or PRP. These results clearly show the significant role of plasma protein in interactions of drugs, including bortezomib, with blood cells, also with platelets. 

Therefore, the paper in the present form, in my opinion, cannot be recommended for publication – major revision is required.

Major comments

Sometimes the Authors used wording that platelets were stimulated with bortezomib (pp. 3, line 120). In my opinion this is not appropriate wording because BTZ can stimulate platelets providing to showing some platelet activation markers but BTZ is not platelet agonist. Additionally, in this study the sentence that BTZ induces platelet activation, was a hypothesis what was verified in experiment.

The abstract should be revised. The most important data as the values and p significance should be included. Also, the conclusions were missed in the present form of abstract.

The Authors quoted the publication of Nayak et al., but also Klingler et al. and Koessler et al. reported very important data about bortezomib and human blood platelets. These publications should be cited in manuscript.

Klingler P, Niklaus M, Koessler J, Weber K, Koessler A, Boeck M, Kobsar A. Influence of long-term proteasome inhibition on platelet responsiveness mediated by bortezomib. Vascul Pharmacol. 2021 Jun;138:106830. doi: 10.1016/j.vph.2021.106830.

Koessler J, Schuepferling A, Klingler P, Koessler A, Weber K, Boeck M, Kobsar A. The role of proteasome activity for activating and inhibitory signalling in human platelets. Cell Signal. 2019 Oct;62:109351. doi: 10.1016/j.cellsig.2019.109351.

Koessler J, Etzel J, Weber K, Boeck M, Kobsar A. Evaluation of dose-dependent effects of the proteasome inhibitor bortezomib in human platelets. Eur J Pharmacol. 2016 Nov 15;791:99-104. doi: 10.1016/j.ejphar.2016.08.031.

The using of means and SEM to showing data on figures is not eligible. It should be rather medians and interquartile ranges because some data were not normally distributed.

Paragraph 4.2.

How many volunteers participated in this study?

Figures 3A and 3B show the same data (%). The Authors may consider changing figures and i.e. may present at figure 3B median of fluorescence intensity for DiOC6(3).

The manuscript needs English correction. Some minor language mistakes should be revised.

Minor comments:

The Authors should consider revision of keywords. In my opinion, the next keywords: proteasome inhibitor, human blood platelet would be better that gel-filtered human platelets.

In my opinion, the first paragraph from the Results should be rather included into Discussion or Introduction section.

The part of sentence „Proteasome inhibitors hinder proteasome activity,…. (pp. 1, line 38) should be changed, this information is obvious.

Thrombin can be used as a blood platelet stimulator not only in isolated platelet suspension but also in PRP or whole blood when GPRP peptide is used (GPRP prevents polymerization of fibrin).

The Discussion section should be revised a little. The sentences from line 230 to 240 (pp. 8) can be moved from Discussion to Introduction section.

The content of BCSG buffer should be removed from the paragraph 4.5.3. (pp. 11) because this information is written in paragraph 4.4. (pp.10).

Author Response

Response to Reviewer 4.

We would like to thank the Reviewer for the valuable comments. We accept the suggestions and carried out the changes in the revised version. Here we also provide a point by point answer to the points raised.

Major comments:

  1. The term ’stimulated’ was deleted when the platelets were incubated with bortezomib. Instead we used the terms ’pretreated’ or ’preincubated’ throughout the manuscript. We also replaced the platelet activation part to the Discussion chapter after data pertaining to platelet activation have been presented.

  1. The word count of the Abstract is limited, however we deleted some parts to include the most important numerical results as well as the conclusions. We agree with the Reviewer that in general ’an abstract needs numbers’.

  1. We are grateful for directing our attention onto the three manuscripts listed. We enclosed all three in the Discussion part of the revised version.

  1. The Reviewer is right that some of our data did not follow Gaussian distribution. Thus, here the relevant figures PS expression on Figure 1 and the depolarized cell percentages on Figure 3A have been redrawn by presenting the median and the box and whisker plots. The ’p ’values for the statistical differences did not change.

  1. Overall 41 volunteers participated in the study.

  1. We recreated Figure 3, partly because of the aforementioned statistical changes and partly as the Reviewer suggested we now replaced the dot plots with the median fluorescence intensity values on Figure 3B and the statistical differences were comparable to that of the depolarized cells.

  1. We asked one of our native English speaking colleague to inspect the manuscript and he made several corrections.

Minor comments:

  1. We accepted the suggested keywords and replaced the ones used previously (see line 26).

  1. The first paragraph of the results has been included into the first paragraph of the discussion (lines 236 to 241).

  1. The part of the sentence that says proteasome inhibitors hinder proteasome activity has been changed in the second paragraph of the introduction, line 41).

  1. I agree with the reviewer that thrombin can be used in PRP or whole blood when GPRP peptide is used. This part of the sentence has been rephrased accordingly (second paragraph of the 2.1 section of the results, line 99).

  1. The discussion section has been revised. The first paragraph of the discussion has been removed entirely in the revised manuscript.

  1. The content of BSGC buffer has been removed from paragraph 4.5.3 as suggested by the reviewer (see line 398).

Round 2

Reviewer 1 Report

Authors addressed all the comments this review had and now the manuscript has a nice scientific soundness. 

Author Response

We thank the reviewer for the effort devoted in our manuscript

Reviewer 2 Report

The authors responded to most comments made and made appropriate additions and changes in the manuscript. This is an interesting story in the platelet field.   

Author Response

(The authors gave the same response as above.)

Reviewer 4 Report

The manuscript in a new revised version can be accepted for publication after minor revision.

Minor comments

In the Abstract, the Authors included the data but it is co clear what the numbers mean (“ PS expression was dose and time-dependently increased in the BTZ-treated groups (37%) as was P-18 selectin (46%)” - 37% or 46% is for which dose and time?

The number of study participants should be included into Materials and Methods par. 4.2.

The information about data presentation should be wrote in the legends for Figures 1 and 3.

Author Response

Response to Reviewer 4

 We would like to thank the Reviewer once again for the quick and positive comments. The following responses have been provided accordingly;

- We have decided to remove the median and mean values of PS expression and P-selectin expression for the highest BTZ concentration from the Abstract as given the 200 word count limit, it would not have been possible to accurately describe all concentrations and values.

- The number of study participants has been included into the Materials and Methods, paragraph. 4.2. (Lines 339- 340) as suggested.

- The information about data presentation has been indicated in the legends for Figures 1 and 3 as suggested by the Reviewer.
